# Trading Computation for Communication: Distributed Stochastic Dual Coordinate Ascent

**Tianbao Yang**
NEC Labs America, Cupertino, CA 95014
`tyang@nec-labs.com`

## Abstract

We present and study a distributed optimization algorithm by employing a stochastic dual coordinate ascent method. Stochastic dual coordinate ascent methods enjoy strong theoretical guarantees and often have better performances than stochastic gradient descent methods in optimizing regularized loss minimization problems. It still lacks of efforts in studying them in a distributed framework. We make a progress along the line by presenting a distributed stochastic dual coordinate ascent algorithm in a star network, with an analysis of the tradeoff between computation and communication. We verify our analysis by experiments on real data sets. Moreover, we compare the proposed algorithm with distributed stochastic gradient descent methods and distributed alternating direction methods of multipliers for optimizing SVMs in the same distributed framework, and observe competitive performances.

## 1 Introduction

In recent years of machine learning applications, the size of data has been observed with an unprecedented growth. In order to efficiently solve large scale machine learning problems with millions of and even billions of data points, it has become popular to take advantage of the computational power of multi-cores in a single machine or multi-machines on a cluster to optimize the problems in a parallel fashion or a distributed fashion [2].

In this paper, we consider the following generic optimization problem arising ubiquitously in supervised machine learning applications:

$$\min_{w \in \mathbb{R}^d} P(w), \text{ where } P(w) = \frac{1}{n} \sum_{i=1}^n \phi(w^\top x_i; y_i) + \lambda g(w), \tag{1}$$

where $w \in \mathbb{R}^d$ denotes the linear predictor to be optimized, $(x_i, y_i), x_i \in \mathbb{R}^d, i = 1, \ldots, n$ denote the instance-label pairs of a set of data points, $\phi(z; y)$ denotes a loss function and $g(w)$ denotes a regularization on the linear predictor. Throughout the paper, we assume the loss function $\phi(z; y)$ is convex w.r.t the first argument and we refer to the problem in (1) as Regularized Loss Minimization (RLM) problem.

The RLM problem has been studied extensively in machine learning, and many efficient sequential algorithms have been developed in the past decades [8, 16, 10]. In this work, we aim to solve the problem in a distributed framework by leveraging the capabilities of tens of hundreds of CPU cores. In contrast to previous works of distributed optimization that are based on either (stochastic) gradient descent (GD and SGD) methods [21, 11] or alternating direction methods of multipliers (ADMM) [2, 23], we motivate our research from the recent advances on (stochastic) dual coordinate ascent (DCA and SDCA) algorithms [8, 16]. It has been observed that DCA and SDCA algorithms can have comparable and sometimes even better convergence speed than GD and SGD methods. However, it lacks efforts in studying them in a distributed fashion and comparing to those SGD-based and ADMM-based distributed algorithms.

In this work, we bridge the gap by developing a Distributed Stochastic Dual Coordinate Ascent (DisDCA) algorithm for solving the RLM problem. We summarize the proposed algorithm and our contributions as follows:

- The presented DisDCA algorithm possesses two key characteristics: (i) parallel computation over $K$ machines (or cores); (ii) sequential updating of $m$ dual variables per iteration on individual machines followed by a "reduce" step for communication among processes. It enjoys a strong guarantee of convergence rates for smooth or no-smooth loss functions.

- We analyze the tradeoff between computation and communication of DisDCA invoked by $m$ and $K$. Intuitively, increasing the number $m$ of dual variables per iteration aims at reducing the number of iterations for convergence and therefore mitigating the pressure caused by communication. Theoretically, our analysis reveals the effective region of $m, K$ versus the regularization path of $\lambda$.

- We present a practical variant of DisDCA and make a comparison with distributed ADMM. We verify our analysis by experiments and demonstrate the effectiveness of DisDCA by comparing with SGD-based and ADMM-based distributed optimization algorithms running in the same distributed framework.

## 2 Related Work

Recent years have seen the great emergence of distributed algorithms for solving machine learning related problems [2, 9]. In this section, we focus our review on distributed optimization techniques. Many of them are based on stochastic gradient descent methods or alternating direction methods of multipliers.

Distributed SGD methods utilize the computing resources of multiple machines to handle a large number of examples simultaneously, which to some extent alleviates the high computational load per iteration of GD methods and also improve the performances of sequential SGD methods. The simplest implementation of a distributed SGD method is to calculate the stochastic gradients on multiple machines, and to collect these stochastic gradients for updating the solution on a master machine. This idea has been implemented in a MapReduce framework [13, 4] and a MPI framework [21, 11]. Many variants of GD methods have be deployed in a similar style [1]. ADMM has been employed for solving machine learning problems in a distributed fashion [2, 23], due to its superior convergences and performances [5, 23]. The original ADMM [7] is proposed for solving equality constrained minimization problems. The algorithms that adopt ADMM for solving the RLM problems in a distributed framework are based on the idea of global variable consensus. Recently, several works [19, 14] have made efforts to extend ADMM to its online or stochastic versions. However, they suffer relatively low convergence rates.

The advances on DCA and SDCA algorithms [12, 8, 16] motivate the present work. These studies have shown that in some regimes (e.g., when a relatively high accurate solution is needed), SDCA can outperform SGD methods. In particular, S. Shalev-Shwartz and T. Zhang [16] have derived new bounds on the duality gap, which have been shown to be superior to earlier results. However, there still lacks of efforts in extending these types of methods to a distributed fashion and comparing them with SGD-based algorithms and ADMM-based distributed algorithms. We bridge this gap by presenting and studying a distributed stochastic dual ascent algorithm. It has been brought to our attention that M. Takác et al. [20] have recently published a paper to study the parallel speedup of mini-batch primal and dual methods for SVM with hinge loss and establish the convergence bounds of mini-batch Pegasos and SDCA depending on the size of the mini-batch. This work differentiates from their work in that (i) we explicitly take into account the tradeoff between computation and communication; (ii) we present a more practical variant and make a comparison between the proposed algorithm and ADMM in view of solving the subproblems, and (iii) we conduct empirical studies for comparison with these algorithms. Other related but different work include [3], which presents Shotgun, a parallel coordinate descent algorithm for solving $\ell_1$ regularized minimization problems.

There are other unique issues arising in distributed optimization, e.g., synchronization vs asynchronization, star network vs arbitrary network. All these issues are related to the tradeoff between communication and computation [22, 24]. Research in these aspects are beyond the scope of this work and can be considered as future work.

# 3 Distributed Stochastic Dual Coordinate Ascent

In this section, we present a distributed stochastic dual coordinate ascent (DisDCA) algorithm and its convergence bound, and analyze the tradeoff between computation and communication. We also present a practical variant of DisDCA and make a comparison with ADMM. We first present some notations and preliminaries.

For simplicity of presentation, we let $\phi_i(w^\top x_i) = \phi(w^\top x_i; y_i)$. Let $\phi_i^*(\alpha)$ and $g^*(v)$ be the convex conjugate of $\phi_i(z)$ and $g(w)$, respectively. We assume $g^*(v)$ is continuous differentiable. It is easy to show that the problem in (1) has a dual problem given below:

$$\max_{\alpha \in \mathbb{R}^n} D(\alpha), \text{ where } D(\alpha) = \frac{1}{n} \sum_{i=1}^{n} -\phi_i^*(-\alpha_i) - \lambda g^* \left( \frac{1}{\lambda n} \sum_{i=1}^{n} \alpha_i x_i \right). \tag{2}$$

Let $w_*$ be the optimal solution to the primal problem in (1) and $\alpha_*$ be the optimal solution to the dual problem in (2). If we define $v(\alpha) = \frac{1}{\lambda n} \sum_{i=1}^{n} \alpha_i x_i$, and $w(\alpha) = \nabla g^*(v)$, it can be verified that $w(\alpha_*) = w_*, P(w(\alpha_*)) = D(\alpha_*)$. In this paper, we aim to optimize the dual problem (2) in a distributed environment where the data are distributed evenly across over $K$ machines. Let $(x_{k,i}, y_{k,i}), i = 1, \ldots, n_k$ denote the training examples on machine $k$. For ease of analysis, we assume $n_k = n/K$. We denote by $\alpha_{k,i}$ the associated dual variable of $x_{k,i}$, and by $\phi_{k,i}(\cdot), \phi_{k,i}^*(\cdot)$ the corresponding loss function and its convex conjugate. To simplify the analysis of our algorithm and without loss of generality, we make the following assumptions about the problem:

- $\phi_i(z)$ is either a $(1/\gamma)$-smooth function or a $L$-Lipschitz continuous function (c.f. the definitions given below). Exemplar smooth loss functions include e.g., $L_2$ hinge loss $\phi_i(z) = \max(0, 1 - y_i z)^2$, logistic loss $\phi_i(z) = \log(1 + \exp(-y_i z))$. Commonly used Lipschitz continuous functions are $L_1$ hinge loss $\phi_i(z) = \max(0, 1 - y_i z)$ and absolute loss $\phi_i(z) = |y_i - z|$.

- $g(w)$ is a 1-strongly convex function w.r.t to $\| \cdot \|_2$. Examples include $\ell_2$ norm square $1/2\|w\|_2^2$ and elastic net $1/2\|w\|_2^2 + \mu\|w\|_1$.

- For all $i$, $\|x_i\|_2 \leq 1$, $\phi_i(z) \geq 0$ and $\phi_i(0) \leq 1$.

**Definition 1.** *A function $\phi(z) : \mathbb{R} \to \mathbb{R}$ is a L-Lipschitz continuous function, if for all $a, b \in \mathbb{R}$ $|\phi(a) - \phi(b)| \leq L|a - b|$. A function $\phi(z) : \mathbb{R} \to \mathbb{R}$ is $(1/\gamma)$-smooth, if it is differentiable and its gradient $\nabla\phi(z)$ is $(1/\gamma)$-Lipschitz continuous, or for all $a, b \in \mathbb{R}$, we have $\phi(a) \leq \phi(b) + (a - b)^\top \nabla\phi(b) + \frac{1}{2\gamma}(a - b)^2$. A convex function $g(w) : \mathbb{R}^d \to \mathbb{R}$ is $\beta$-strongly convex w.r.t a norm $\| \cdot \|$, if for any $s \in [0, 1]$ and $w_1, w_2 \in \mathbb{R}^d$, $g(sw_1 + (1 - s)w_2) \leq sg(w_1) + (1 - s)g(w_2) - \frac{1}{2}s(1 - s)\beta\|w_1 - w_2\|^2$.*

## 3.1 DisDCA Algorithm: The Basic Variant

The detailed steps of the basic variant of the **DisDCA** algorithm are described by a pseudo code in Figure 1. The algorithm deploys $K$ processes running simultaneously on $K$ machines (or cores)[1], each of which only accesses its associated training examples. Each machine calls the same procedure **SDCA-mR**, where **mR** manifests two unique characteristics of SDCA-mR compared to SDCA. (i) At each iteration of the outer loop, $m$ examples instead of one are randomly sampled for updating their dual variables. This is implemented by an inner loop that costs the most computation at each outer iteration. (ii) After updating the $m$ randomly selected dual variables, it invokes a function **Reduce** to collect the updated information from all $K$ machines that accommodates naturally to the distributed environment. The **Reduce** function acts exactly like MPI::AllReduce if one wants to implement the algorithm in a MPI framework. It essentially sends $\Delta v_k = \frac{1}{\lambda n} \sum_{j=1}^{m} \Delta\alpha_{k,i_j} x_{i_j}$ to a process, adds all of them to $v^{t-1}$, and then broadcasts the updated $v^t$ to all $K$ processes. It is this step that involves the communication among the $K$ machines. Intuitively, smaller $m$ yields less computation and slower convergence and therefore more communication and vice versa. In next subsection, we would give a rigorous analysis about the convergence, computation and communication.

**Remark:** The goal of the updates is to increase the dual objective. The particular options presented in routine **IncDual** is to maximize the lower bounds of the dual objective. More options are provided

<div style="border:1px solid">

**DisDCA Algorithm (The Basic Variant)**

Start $K$ processes by calling the following procedure **SDCA-mR** with input $m$ and $T$

Procedure **SDCA-mR**

**Input:** number of iterations $T$, number of samples $m$ at each iteration

**Let:** $\alpha_k^0 = 0, v^0 = 0, w^0 = \nabla g^*(0)$

**Read Data:** $(x_{k,i}, y_{k,i}), i = 1, \cdots, n_k$

**Iterate:** for $t = 1, \ldots, T$

    **Iterate:** for $j = 1, \ldots, m$

      Randomly pick $i \in \{1, \cdots, n_k\}$ and let $i_j = i$

      Find $\Delta\alpha_{k,i}$ by calling routine **IncDual**$(w = w^{t-1}, scl = mK)$

      Set $\alpha_{k,i}^t = \alpha_{k,i}^{t-1} + \Delta\alpha_{k,i}$

    **Reduce**: $v^t : \frac{1}{\lambda n}\sum_{j=1}^m \Delta\alpha_{k,i_j}x_{k,i_j} \rightarrow v^{t-1}$

    **Update**: $w^t = \nabla g^*(v^t)$

---

Routine **IncDual**$(w, scl)$

Option I:

  Let $\Delta\alpha_{k,i} = \max_{\Delta\alpha} -\phi_{k,i}^*(-(\alpha_{k,i}^{t-1} + \Delta\alpha)) - \Delta\alpha x_{k,i}^\top w - \dfrac{scl}{2\lambda n}(\Delta\alpha)^2\|x_{k,i}\|_2^2$

Option II:

  Let $z_{k,i}^{t-1} = -\partial\phi_{k,i}(x_{k,i}^\top w) - \alpha_{k,i}^{t-1}$

  Let $\Delta\alpha_{k,i} = s_{k,i}z_{k,i}^{t-1}$ where $s_{k,i} \in [0,1]$ maximize

  $s(\phi_{k,i}^*(-\alpha_{k,i}^{t-1}) + \phi_{k,i}(x_{k,i}^\top w^{t-1}) + z_{k,i}^{t-1}x_{k,i}^\top w) + \dfrac{\gamma s(1-s)}{2}(z_{k,i}^{t-1})^2 - \dfrac{scl}{2\lambda n}s^2(z_{k,i}^{t-1})^2\|x_{k,i}\|_2^2$

</div>

Figure 1: The Basic Variant of the DisDCA Algorithm

in supplementary materials. The solutions to option I have closed forms for several loss functions (e.g., $L_1, L_2$ hinge losses, square loss and absolute loss) [16]. Note that different from the options presented in [16], the ones in **Incdual** use a slightly different scalar factor $mK$ in the quadratic term to adapt for the number of updated dual variables.

### 3.2 Convergence Analysis: Tradeoff between Computation and Communication

In this subsection, we present the convergence bound of the DisDCA algorithm and analyze the tradeoff between computation, convergence or communication. The theorem below states the convergence rate of DisDCA algorithm for smooth loss functions (The omitted proofs and other derivations can be found in supplementary materials) .

**Theorem 1.** *For a $(1/\gamma)$-smooth loss function $\phi_i$ and a 1-strongly convex function $g(w)$, to obtain an $\epsilon_p$ duality gap of $\mathrm{E}[P(w^T) - D(\alpha^T)] \leq \epsilon_P$, it suffices to have*

$$T \geq \left(\frac{n}{mK} + \frac{1}{\lambda\gamma}\right)\log\left(\left(\frac{n}{mK} + \frac{1}{\lambda\gamma}\right)\frac{1}{\epsilon_P}\right).$$

**Remark:** In [20], the authors established a convergence bound of mini-batch SDCA for $L_1$-SVM that depends on the spectral norm of the data. Applying their trick to our algorithmic framework is equivalent to replacing the scalar $mK$ in DisDCA algorithm with $\beta_{mK}$ that characterizes the spectral norm of sampled data across all machines $X_{mK} = (x_{11}, \ldots x_{1m}, \ldots, x_{Km})$. The resulting convergence bound for $(1/\gamma)$-smooth loss functions is given by substituting the term $\frac{1}{\lambda\gamma}$ with $\frac{\beta_{mK}}{mK}\frac{1}{\lambda\gamma}$. The value of $\beta_{mK}$ is usually smaller than $mK$ and the authors in [20] have provided an expression for computing $\beta_{mK}$ based on the spectral norm $\sigma$ of the data matrix $X/\sqrt{n} = (x_1, \ldots x_n)/\sqrt{n}$. However, in practice the value of $\sigma$ cannot be computed exactly. A safe upper bound of $\sigma = 1$ assuming $\|x_i\|_2 \leq 1$ gives the value $mK$ to $\beta_{mK}$, which reduces to the scalar as presented in Figure 1. The authors in [20] also presented an aggressive variant to adjust $\beta$ adaptively and observed improvements. In Section 3.3 we develop a practical variant that enjoys more speed-up compared to the basic variant and their aggressive variant.

**Tradeoff between Computation and Communication** We are now ready to discuss the tradeoff between computation and communication based on the worst case analysis as indicated by Theo-

rem 1. For the analysis of tradeoff between computation and communication invoked by the number of samples $m$ and the number of machines $K$, we fix the number of examples $n$ and the number of dimensions $d$. When we analyze the tradeoff involving $m$, we fix $K$ and vice versa. In the following analysis, we assume the size of model to be communicated is fixed $d$ and is independent of $m$, though in some cases (e.g., high dimensional sparse data) one may communicate a smaller size of data that depends on $m$.

It is notable that in the bound of the number of iterations, there is a term $1/(\lambda\gamma)$. To take this term into account, we first consider an interesting region of $\lambda$ for achieving a good generalization error. Several pieces of works [17, 18, 6] have suggested that in order to obtain an optimal generalization error, the optimal $\lambda$ scales like $\Theta(n^{-1/(1+\tau)})$, where $\tau \in (0,1]$. For example, the analysis in [18] suggests $\lambda = \Theta\left(\frac{1}{\sqrt{n}}\right)$ for SVM.

First, we consider the tradeoff involving the number of samples $m$ by fixing the number processes $K$. We note that the communication cost is proportional to the number of iterations $T = \Omega\left(\frac{n}{mK} + \frac{n^{1/(1+\tau)}}{\gamma}\right)$, while the computation cost per node is proportional to $mT = \Omega\left(\frac{n}{K} + \frac{mn^{1/(1+\tau)}}{\gamma}\right)$ due to that each iteration involves $m$ examples. When $m \leq \Theta\left(\frac{n^{\tau/(1+\tau)}}{K}\right)$, the communication cost decreases as $m$ increases, and the computation costs increases as $m$ increases, though it is dominated by $\Omega(n/K)$. When the value of $m$ is greater than $\Theta\left(\frac{n^{\tau/(1+\tau)}}{K}\right)$, the communication cost is dominated by $\Omega\left(\frac{n^{1/(1+\tau)}}{\gamma}\right)$, then increasing the value of $m$ will become less influential on reducing the communication cost; while the computation cost would blow up substantially.

Similarly, we can also understand how the number of nodes $K$ affects the tradeoff between the communication cost, proportional to $\tilde{\Omega}(KT) = \tilde{\Omega}\left(\frac{n}{m} + \frac{Kn^{1/(1+\tau)}}{\gamma}\right)$ [2], and the computation cost, proportional to $\Omega\left(\frac{n}{K} + \frac{mn^{1/(1+\tau)}}{\gamma}\right)$. When $K \leq \Theta\left(\frac{n^{\tau/(1+\tau)}}{m}\right)$, as $K$ increases the computation cost would decrease and the communication cost would increase. When it is greater than $\Theta\left(\frac{n^{\tau/(1+\tau)}}{m}\right)$, the computation cost would be dominated by $\Omega\left(\frac{mn^{1/(1+\tau)}}{\gamma}\right)$ and the effect of increasing $K$ on reducing the computation cost would diminish.

According to the above analysis, we can conclude that when $mK \leq \Theta(n\lambda\gamma)$, to which we refer as the effective region of $m$ and $K$, the communication cost can be reduced by increasing the number of samples $m$ and the computation cost can be reduced by increasing the number of nodes $K$. Meanwhile, increasing the number of samples $m$ would increase the computation cost and similarly increasing the number of nodes $K$ would increase the communication cost. It is notable that the larger the value of $\lambda$ the wider the effective region of $m$ and $K$, and vice versa. To verify the tradeoff of communication and computation, we present empirical studies in Section 4. Although the smooth loss functions are the most interesting, we present in the theorem below about the convergence of DisDCA for Lipschitz continuous loss functions.

**Theorem 2.** *For a $L$-Lipschitz continuous loss function $\phi_i$ and a $1$-strongly convex function $g(w)$, to obtain an $\epsilon_P$ duality gap $\mathrm{E}[P(\bar{w}_T) - D(\bar{\alpha}_T)] \leq \epsilon_P$, it suffices to have*

$$T \geq \frac{4L^2}{\lambda\epsilon_P} + T_0 + \frac{n}{mK} \geq \frac{20L^2}{\lambda\epsilon_P} + \max\left(0, \frac{n}{mK}\log\left(\frac{\lambda n}{2mKL^2}\right)\right) + \frac{n}{mK},$$

*where $\bar{w}_T = \sum_{t=T_0}^{T-1} w^t/(T-T_0), \bar{\alpha}_T = \sum_{t=T_0}^{T-1} \alpha^t/(T-T_0)$.*

**Remark:** In this case, the effective region of $m$ and $K$ is $mK \leq \Theta(n\lambda\epsilon_P)$ which is narrower than that for smooth loss functions, especially when $\epsilon_P \ll \gamma$. Similarly, if one can obtain an accurate estimate of the spectral norm of all data and use $\beta_{mK}$ in place of $mK$ in Figure 1, the convergence bound can be improved with $\frac{4L^2}{\lambda\epsilon_P}\frac{\beta_{mK}}{mK}$ in place of $\frac{4L^2}{\lambda\epsilon_P}$. Again, the practical variant presented in next section yields more speed-up.

<div style="border:1px solid">

**the practical updates at the $t$-th iteration**

**Initialize:** $u_t^0 = w^{t-1}$
**Iterate:** for $j = 1, \ldots, m$
  Randomly pick $i \in \{1, \cdots, n_k\}$ and let $i_j = i$
  Find $\Delta\alpha_{k,i}$ by calling routine **IncDual**($w = u_t^{j-1}$, $scl = k$)
  Update $\alpha_{k,i}^t = \alpha_{k,i}^{t-1} + \Delta\alpha_{k,i}$ and update $u_t^j = u_t^{j-1} + \frac{1}{\lambda n_k}\Delta\alpha_{k,i}x_{k,i}$

</div>

Figure 2: the updates at the $t$-th iteration of the practical variant of DisDCA

### 3.3 A Practical Variant of DisDCA and A Comparison with ADMM

In this section, we first present a practical variant of DisDCA motivated by intuition and then we make a comparison between DisDCA and ADMM, which provides us more insight about the practical variant of DisDCA and differences between the two algorithms. In what follows, we are particularly interested in $\ell_2$ norm regularization where $g(w) = \|w\|_2^2/2$ and $v = w$.

**A Practical Variant** We note that in Algorithm 1, when updating the values of the following sampled dual variables, the algorithm does not use the updated information but instead $w^{t-1}$ from last iteration. Therefore a potential improvement would be leveraging the up-to-date information for updating the dual variables. To this end, we maintain a local copy of $w_k$ in each machine. At the beginning of the iteration $t$, all $w_k^0, k = 1, \cdots, K$ are synchronized with the global $w^{t-1}$. Then in individual machines, the $j$-th sampled dual variable is updated by **IncDual**($w_k^{j-1}, k$) and the local copy $w_k^j$ is also updated by $w_k^j = w_k^{j-1} + \frac{1}{\lambda n_k}\Delta\alpha_{k,i_j}x_{k,i_j}$ for updating the next dual variable. At the end of the iteration, the local solutions are synchronized to the global variable $w^t = w^{t-1} + \frac{1}{\lambda n}\sum_{k=1}^K \sum_{j=1}^m \Delta\alpha_{k,i_j}^t x_{k,i_j}$. It is important to note that the scalar factor in **IncDual** is now $k$ because the dual variables are updated incrementally and there are $k$ processes running parallell. The detailed steps are presented in Figure 2, where we abuse the same notation $u_t^j$ for the local variable at all processes. The experiments in Section 4 verify the improvements of the practical variant vs the basic variant. It still remains an open problem to us what is the convergence bound of this practical variant. However, next we establish a connection between DisDCA and ADMM that sheds light on the motivation behind the practical variant and the differences between the two algorithms.

**A Comparison with ADMM** First we note that the goal of the updates at each iteration in DisDCA is to increase the dual objective by maximizing the following objective:

$$\max_\alpha \frac{1}{n_k}\sum_{i=1}^m -\phi_i^*(-\alpha_i) - \frac{\lambda}{2}\left\|\hat{w}^{t-1} + 1/(\lambda n_k)\sum_{i=1}^m \alpha_i x_i\right\|_2^2, \tag{3}$$

where $\hat{w}^{t-1} = w^{t-1} - 1/(\lambda n_k)\sum_{i=1}^m \alpha_i^{t-1}x_i$ and we suppress the subscript $k$ associated with each machine. The updates presented in Algorithm 1 are solutions to maximizing the lower bounds of the above objective function by decoupling the $m$ dual variables. It is not difficult to derive that the dual problem in (3) has the following primal problem (a detailed derivation and others can be found in supplementary materials):

$$\text{DisDCA:}\quad \min_w \frac{1}{n_k}\sum_{i=1}^m \phi_i(x_i^\top w) + \frac{\lambda}{2}\left\|w - \left(w^{t-1} - 1/(\lambda n_k)\sum_{i=1}^m \alpha_i^{t-1}x_i\right)\right\|_2^2. \tag{4}$$

We refer to $\hat{w}^t$ as the penalty solution. Second let us recall the updating scheme in ADMM. The (deterministic) ADMM algorithm at iteration $t$ solves the following problems in each machine:

$$\text{ADMM:}\quad w_k^t = \arg\min_w \frac{1}{n_k}\sum_{i=1}^{n_k} \phi_i(x_i^\top w) + \frac{\rho K}{2}\|w - \underbrace{(w^{t-1} - u_k^{t-1})}_{\hat{w}^{t-1}}\|_2^2, \tag{5}$$

where $\rho$ is a penalty parameter and $w^{t-1}$ is the global primal variable updated by

$$w^t = \frac{\rho K(\bar{w}^t + \bar{u}^{t-1})}{\rho K + \lambda}, \text{ with } \bar{w}^t = \frac{1}{K}\sum_{k=1}^K w_k^t, \ \bar{u}^{t-1} = \frac{1}{K}\sum_{k=1}^K u_k^{t-1},$$

and $u_k^{t-1}$ is the local "dual" variable updated by $u_k^t = u_k^{t-1} + w_k^t - w^t$. Comparing the subproblem (4) in DisDCA and the subproblem (5) in ADMM leads to the following observations. (1) Both aim at solving the same type of problem to increase the dual objective or decrease the primal objective. DisDCA uses only $m$ randomly selected examples while ADMM uses all examples. (2) However, the penalty solution $\hat{w}^{t-1}$ and the penalty parameter are different. In DisDCA, $\hat{w}^{t-1}$ is constructed by subtracting from the global solution the local solution defined by the dual variables $\alpha$, while in ADMM it is constructed by subtracting from the global solution the local Lagrangian variables $u$. The penalty parameter in DisDCA is given by the regularization parameter $\lambda$ while that in ADMM is a parameter that is needed to be specified by the user.

Now, let us explain the practical variant of DisDCA from the viewpoint of inexactly solving the subproblem (4). Note that if the optimal solution to (3) is denoted by $\alpha_i^*, i = 1, \ldots, m$, then the optimal solution $u^*$ to (4) is given by $u^* = \hat{w}^{t-1} + \frac{1}{\lambda n_k} \sum_{i=1}^m \alpha_i^* x_i$. In fact, the updates at the $t$-th iteration of the practical variant of DisDCA is to optimize the subproblem (4) by the SDCA algorithm with only **one** pass of the sampled data points and an **initialization** of $\alpha_i^0 = \alpha_i^{t-1}, i = 1 \ldots, m$. It means that the initial primal solution for solving the subproblem (3) is $u^0 = \hat{w}^{t-1} + \frac{1}{\lambda n_k} \sum_{i=1}^m \alpha_i^{t-1} x_i = w^{t-1}$. That explains the initialization step in Figure 2.

In a recent work [23] of applying ADMM to solving the $L_2$-SVM problem in the same distributed fashion, the authors exploited different strategies for solving the subproblem (5) associated with $L_2$-SVM, among which the DCA algorithm with only one pass of all data points gives the best performance in terms of running time (e.g., it is better than DCA with several passes of all data points and is also better than a trusted region Newton method). This from another point of view validates the practical variant of DisDCA.

Finally, it is worth to mention that unlike ADMM whose performance is significantly affected by the value of the penalty parameter $\rho$, DisDCA is a parameter free algorithm.

## 4   Experiments

In this section, we present some experimental results to verify the theoretical results and the empirical performances of the proposed algorithms. We implement the algorithms by C++ and openMPI and run them in cluster. On each machine, we only launch one process. The experiments are performed on two large data sets with different number of features, covtype and kdd. Covtype data has a total of $581,012$ examples and $54$ features. Kdd data is a large data used in kdd cup 2010, which contains $19,264,097$ training examples and $29,890,095$ features. For covtype data, we use $522,911$ examples for training. We apply the algorithms to solving two SVM formulations, namely $L_2$-SVM with hinge loss square and $L_1$-SVM with hinge loss, to demonstrate the capabilities of DisDCA in solving smooth loss functions and Lipschitz continuous loss functions. In the legend of figures, we use DisDCA-b to denote the basic variant, DisDCA-p to denote the practical variant, and DisDCA-a to denote the aggressive variant of DisDCA [20].

**Tradeoff between Communication and Computation** To verify the convergence analysis, we show in Figures 3(a)∼3(b), 3(d)∼3(e) the duality gap of the basic variant and the practical variant of the DisDCA algorithm versus the number of iterations by varying the number of samples $m$ per iteration, the number of machines $K$ and the values of $\lambda$. The results verify the convergence bound in Theorem 1. At the beginning of increasing the values of $m$ or $K$, the performances are improved. However, when their values exceed certain number, the impact of increasing $m$ or $K$ diminishes. Additionally, the larger the value of $\lambda$ the wider the effective region of $m$ and $K$. It is notable that the effective region of $m$ and $K$ of the practical variant is much larger than that of the basic variant. We also briefly report a running time result: to obtain an $\epsilon = 10^{-3}$ duality gap for optimizing $L_2$-SVM on covtype data with $\lambda = 10^{-3}$, the running time of DisDCA-p with $m = 1, 10, 10^2, 10^3$ by fixing $K = 10$ are $30, 4, 0, 5$ seconds [3], respectively, and the running time with $K = 1, 5, 10, 20$ by fixing $m = 100$ are $3, 0, 0, 1$ seconds, respectively. The speed-up gain on kdd data by increasing $m$ is even larger because the communication cost is much higher. In supplement, we present more results on visualizing the communication and computation tradeoff.

**The Practical Variant vs The Basic Variant** To further demonstrate the usefulness of the practical variant, we present a comparison between the practical variant and the basic variant for optimizing

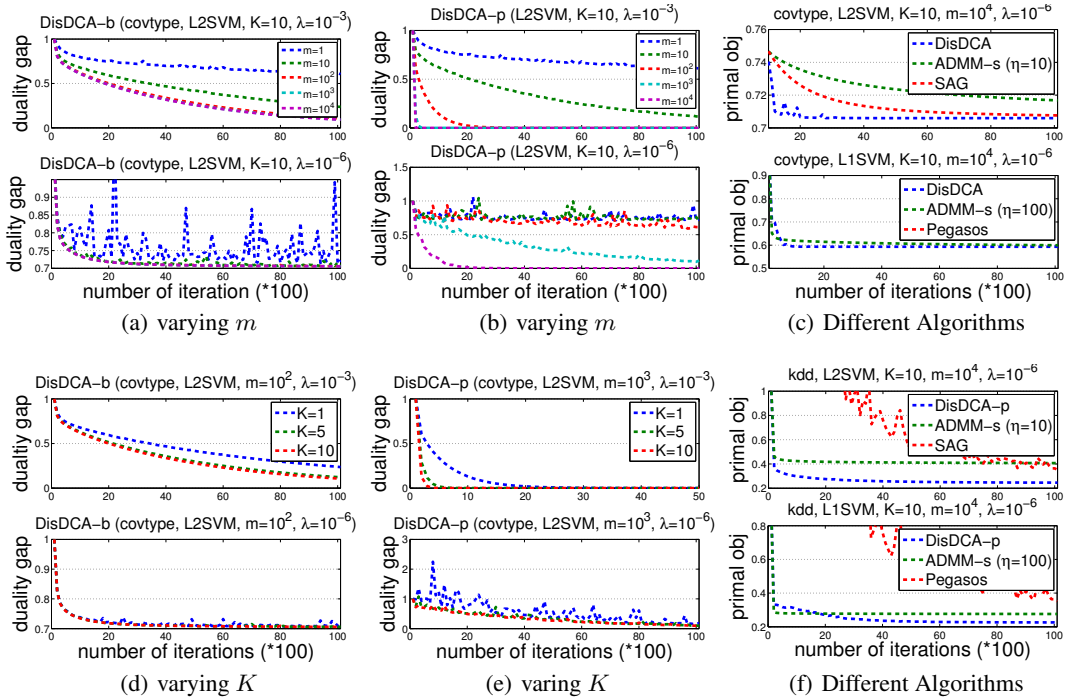

Figure 3: (a,b): duality gap with varying $m$; (d,e): duality gap with varying $K$; (c, f) comparison of different algorithms for optimizing SVMs. More results can be found in supplementary materials.

the two SVM formulations in supplementary material. We also include the performances of the aggressive variant proposed in [20], by applying the aggressive updates on the $m$ sampled examples in each machine without incurring additional communication cost. The results show that the practical variant converges much faster than the basic variant and the aggressive variant.

**Comparison with other baselines** Lastly, we compare DisDCA with SGD-based and ADMM-based distributed algorithms running in the same distributed framework. For optimizing $L_2$-SVM, we implement the stochastic average gradient (SAG) algorithm [15], which also enjoys a linear convergence for smooth and strongly convex problems. We use the constant step size $(1/L_s)$ suggested by the authors for obtaining a good practical performance, where the $L_s$ denotes the smoothness parameter of the problem, set to $2R + \lambda$ given $\|x_i\|_2^2 \leq R, \forall i$. For optimizing $L_1$-SVM, we compare to the stochastic Pegasos. For ADMM-based algorithms, we implement a stochastic ADMM in [14] (ADMM-s) and a deterministic ADMM in [23] (ADMM-dca) that employes the DCA algorithm for solving the subproblems. In the stochastic ADMM, there is a step size parameter $\eta_t \propto 1/\sqrt{t}$. We choose the best initial step size among $[10^{-3}, 10^3]$. We run all algorithms on $K = 10$ machines and set $m = 10^4, \lambda = 10^{-6}$ for all stochastic algorithms. In terms of the parameter $\rho$ in ADMM, we find that $\rho = 10^{-6}$ yields good performances by searching over a range of values. We compare DisDCA with SAG, Pegasos and ADMM-s in Figures 3(c), 3(f) [4], which clearly demonstrate that DisDCA is a strong competitor in optimizing SVMs. In supplement we compare DisDCA by setting $m = n_k$ against ADMM-dca with four different values of $\rho = 10^{-6}, 10^{-4}, 10^{-2}, 1$ on kdd. The results show that the performances deteriorate significantly if the $\rho$ is not appropriately set, while DisDCA can produce comparable performance without additional efforts in tuning the parameter.

## 5 Conclusions

We have presented a distributed stochastic dual coordinate descent algorithm and its convergence rates, and analyzed the tradeoff between computation and communication. The practical variant has substantial improvements over the basic variant and other variants. We also make a comparison with other distributed algorithms and observe competitive performances.

## Footnotes

[1]We use process and machine interchangeably.

[2]We simply ignore the communication delay in our analysis.

[3] 0 second means less than 1 second. We exclude the time for computing the duality gap at each iteration.

[4]The primal objective of Pegasos on covtype is above the display range.

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
