[Supplementary Material]

# Supplementary Materials:
# Trading Computation for Communication:
# Distributed Stochastic Dual Coordinate Ascent

**Tianbao Yang**
NEC Labs America, Cupertino, CA 95014
tyang@nec-labs.com

## 1 Proof of Theorem 1 and Theorem 2

For the proof of Theorem 1, we first prove the following Lemma.

**Lemma 1.** *Assume that $\phi_i^*(z)$ is $\gamma$-strongly convex function (where $\gamma$ can be zero). Then for any $t > 0$ and $s \in [0, 1]$, we have*

$$\mathrm{E}\left[D(\alpha^t) - D(\alpha^{t-1})\right] \geq \frac{smK}{n}\mathrm{E}[P(w^{t-1}) - D(\alpha^{t-1})] - \left(\frac{smK}{n}\right)^2 \frac{G^t}{2\lambda}$$

*where*

$$G^t = \frac{1}{n}\sum_{i=1}^{n}\left(1 - \frac{\gamma(1-s)\lambda n}{smK}\right)(u_i^{t-1} - \alpha_i^{t-1})^2,$$

*and $u_i^{t-1} = -\nabla\phi_i(x_i^\top w^{t-1})$.*

*Proof of Lemma 1.* For simplicity, we let $\mathcal{I}_m^k$ denote the random index $\{i_1, \cdots, i_m\}$ randomly sampled at iteration $t$ in machine $k$. We begin by bounding the improvement in the dual objective. By the definition of $D(\alpha)$, we have

$$n[D(\alpha^t) - D(\alpha^{t-1})] = \underbrace{\left[\sum_{k=1}^{K}\sum_{i\in\mathcal{I}_m^k}-\phi_{k,i}^*(-\alpha_{k,i}^t) - \lambda n g^*\left(v^t\right)\right]}_{A} - \underbrace{\left[\sum_{k=1}^{K}\sum_{i\in\mathcal{I}_m^k}-\phi_{k,i}(\alpha_{k,i}^{t-1}) - \lambda n g^*\left(v^{t-1}\right)\right]}_{B}$$

By the definition of the update, we have

$$g^*(v^t) = g^*\left(v^{t-1} + \frac{1}{\lambda n}\sum_{k=1}^{K}\sum_{i\in\mathcal{I}_m^k}\Delta\alpha_{k,i}^t x_{k,i}\right)$$

$$\leq g^*(v^{t-1}) + \frac{1}{\lambda n}\sum_{k=1}^{K}\sum_{i\in\mathcal{I}_m^k}\Delta\alpha_{k,i}^t x_{k,i}^\top\nabla g^*(v^{t-1}) + \frac{1}{2}\left\|\frac{1}{\lambda n}\sum_{k=1}^{K}\sum_{i\in\mathcal{I}_m^k}\Delta\alpha_i^t x_i\right\|_2^2$$

$$= g^*(v^{t-1}) + \frac{1}{\lambda n}\sum_{k=1}^{K}\sum_{i\in\mathcal{I}_m^k}\Delta\alpha_{k,i}^t x_{k.i}^\top w^{t-1} + \frac{1}{2}\left\|\frac{1}{\lambda n}\sum_{k=1}^{K}\sum_{i\in\mathcal{I}_m^k}\Delta\alpha_i^t x_i\right\|_2^2$$

$$\leq g^*(v^{t-1}) + \frac{1}{\lambda n}\sum_{k=1}^{K}\sum_{i\in\mathcal{I}_m^k}\Delta\alpha_{k,i}^t x_{k.i}^\top w^{t-1} + \frac{mK}{2(\lambda n)^2}\sum_{k=1}^{K}\sum_{i\in\mathcal{I}_m^k}(\Delta\alpha_i^t)^2$$

where the first inequality uses the strong convexity of $g^*$, the second equality uses the fact $w^{t-1} = \nabla g^*(v^{t-1})$, and the last inequality uses the Cauchy-Schwarz inequality. Then we can bound $A$ by

$$A \geq \sum_{k=1}^{K} \sum_{i \in \mathcal{I}_m^k} -\phi_{k,i}^*(-(\alpha_{k,i}^{t-1} + \Delta \alpha_{k,i}^t)) - \lambda n g^*(v^{t-1}) - \sum_{k=1}^{K} \sum_{i \in \mathcal{I}_m^k} \Delta \alpha_{k,i}^t x_{k,i}^\top w^{t-1} - \frac{mK}{2\lambda n} \sum_{k=1}^{K} \sum_{i \in \mathcal{I}_m^k} (\Delta \alpha_{k,i}^t)^2$$

We can further bound $A$ as follows if we maximize R.H.S of above inequality over $\Delta \alpha_{k,i}$ or if we restrict $\Delta \alpha_{k,i} = s_{k,i}(u_{k,i}^{t-1} - \alpha_{k,i}^{t-1})$, where $u_{k,i}^{t-1} = -\nabla \phi_{k,i}(x_{k,i}^\top w^{t-1})$.

$$
\begin{aligned}
A \;\geq\; & \sum_{k=1}^{K} \sum_{i \in \mathcal{I}_m^k} -\phi_{k,i}^* \left( -\alpha_{k,i}^{t-1} - s_{k,i}(u_{k,i}^{t-1} - \alpha_{k,i}^{t-1}) \right) - \lambda n g^*(v^{t-1}) \\
& - \sum_{k=1}^{K} \sum_{i \in \mathcal{I}_m^k} s_{k,i}(u_{k,i}^{t-1} - \alpha_i^{t-1}) x_{k,i}^\top w^{t-1} - \frac{mK}{2\lambda n} \sum_{k=1}^{K} \sum_{i \in \mathcal{I}_m^k} s_{k,i}^2 (u_{k,i}^{t-1} - \alpha_{k,i}^{t-1})^2 \\
\geq\; & \sum_{k=1}^{K} \sum_{i \in \mathcal{I}_m^k} \left[ -\phi_{k,i}^*(-\alpha_{k,i}^{t-1}) - s_{k,i}\phi_{k,i}^*(-u_{k,i}^{t-1}) + \frac{\gamma}{2} s_{k,i}(1 - s_{k,i})(u_{k,i}^{t-1} - \alpha_{k,i}^{t-1})^2 \right] - \lambda n g^*(v^{t-1}) \\
& - \sum_{k=1}^{K} \sum_{i \in \mathcal{I}_m^k} s_{k,i}(u_{k,i}^{t-1} - \alpha_{k,i}^{t-1}) x_{k,i}^\top w^{t-1} - \frac{mK}{2\lambda n} \sum_{k=1}^{K} \sum_{i \in \mathcal{I}_m^k} s_{k,i}^2 (u_{k,i}^{t-1} - \alpha_{k,i}^{t-1})^2
\end{aligned}
$$

where in the first inequality, we use the strong convexity of $\phi_{k,i}^*$. Since $s_{k,i}$ are maximized over the R.H.S of above inequalities, then we have for any $s \in [0,1]$

$$
\begin{aligned}
A \;\geq\; & \sum_{k=1}^{K} \sum_{i \in \mathcal{I}_m^k} \left[ -\phi_{k,i}^*(-\alpha_{k,i}^{t-1}) - s\phi_{k,i}^*(-u_{k,i}^{t-1}) + \frac{\gamma}{2} s(1 - s)(u_{k,i}^{t-1} - \alpha_{k,i}^{t-1})^2 \right] - \lambda n g^*(v^{t-1}) \\
& - \sum_{k=1}^{K} \sum_{i \in \mathcal{I}_m^k} s(u_{k,i}^{t-1} - \alpha_{k,i}^{t-1}) x_{k,i}^\top w^{t-1} - \frac{mK}{2\lambda n} \sum_{k=1}^{K} \sum_{i \in \mathcal{I}_m^k} s^2 (u_{k,i}^{t-1} - \alpha_{k,i}^{t-1})^2 \\
=\; & \sum_{k=1}^{K} \sum_{i \in \mathcal{I}_m^k} \Bigg[ \underbrace{-s(\phi_{k,i}^*(-u_{k,i}^{t-1}) + x_{k,i}^\top w^{t-1} u_{k,i}^{t-1})}_{s\phi_{k,i}(x_{k,i}^\top w^{t-1})} \Bigg] + \underbrace{\sum_{k=1}^{K} \sum_{i \in \mathcal{I}_m^k} -\phi_{k,i}^*(-\alpha_{k,i}^{t-1}) - \lambda n g^*(v^{t-1})}_{B} \\
& + \frac{s}{2}\left(\gamma(1-s) - \frac{smK}{\lambda n}\right) \sum_{k=1}^{K} \sum_{i \in \mathcal{I}_m^k} (u_{k,i}^{t-1} - \alpha_{k,i}^{t-1})^2 + s \sum_{k=1}^{K} \sum_{i \in \mathcal{I}_m^k} \left[ \phi_{k,i}^*(-\alpha_{k,i}^{t-1}) + \alpha_{k,i}^{t-1} x_{k,i}^\top w^{t-1} \right] \\
=\; & s \sum_{k=1}^{K} \sum_{i \in \mathcal{I}_m^k} \left[ \phi_{k,i}(x_{k,i}^\top w^{t-1}) + \phi_{k,i}^*(-\alpha_{k,i}^{t-1}) + \alpha_{k,i}^{t-1} x_{k,i}^\top w^{t-1} \right] + B \\
& + \frac{s}{2}\left(\gamma(1-s) - \frac{smK}{\lambda n}\right) \sum_{k=1}^{K} \sum_{i \in \mathcal{I}_m^k} (u_{k,i}^{t-1} - \alpha_{k,i}^{t-1})^2
\end{aligned}
$$

Thus, we have

$$
\begin{aligned}
\frac{A - B}{s} \;\geq\; & \sum_{k=1}^{K} \sum_{i \in \mathcal{I}_m^k} \left[ \phi_{k,i}(x_{k,i}^\top w^{t-1}) + \phi_{k,i}^*(-\alpha_{k,i}^{t-1}) + \alpha_{k,i}^{t-1} x_{k,i}^\top w^{t-1} \right] \\
& - \frac{s}{2\lambda}\left( \frac{mK}{n} - \frac{\gamma(1-s)\lambda}{s} \right) \sum_{k=1}^{K} \sum_{i \in \mathcal{I}_m^k} (u_{k,i}^{t-1} - \alpha_{k,i}^{t-1})^2
\end{aligned}
$$

<div style="border:1px solid">

Routine **IncDual**$(w, scl)$

Option I:

Let $\Delta\alpha_{k,i} = \max\limits_{\Delta\alpha} -\phi_{k,i}^*(-(\alpha_{k,i}^{t-1} + \Delta\alpha)) - \Delta\alpha x_{k,i}^\top w - \dfrac{scl}{2\lambda n}(\Delta\alpha)^2\|x_{k,i}\|_2^2$

Option II:

Let $z_{k,i}^{t-1} = -\partial\phi_{k,i}(x_{k,i}^\top w) - \alpha_{k,i}^{t-1}$

Let $s_{k,i} = \max\limits_{s\in[0,1]} -\phi_{k,i}^*\left(-\alpha_{k,i}^{t-1} - sz_{k,i}^{t-1}\right) - sz_{k,i}^{t-1}x_{k,i}^\top w^{t-1} - \dfrac{scl}{2\lambda n}s^2(z_{k,i}^{t-1})^2$

Let $\Delta\alpha_{k,i} = s_{k,i}z_{k,i}^{t-1}$

Option III:

Let $z_{k,i}^{t-1} = -\partial\phi_{k,i}(x_{k,i}^\top w) - \alpha_{k,i}^{t-1}$

Let $\Delta\alpha_{k,i} = s_{k,i}z_{k,i}^{t-1}$ where $s_{k,i} \in [0,1]$ maximize

$$s(\phi_{k,i}^*(-\alpha_{k,i}^{t-1}) + \phi_{k,i}(x_{k,i}^\top w^{t-1}) + z_{k,i}^{t-1}x_{k,i}^\top w) + \dfrac{\gamma s(1-s)}{2}(z_{k,i}^{t-1})^2 - \dfrac{scl}{2\lambda n}s^2(z_{k,i}^{t-1})^2\|x_{k,i}\|_2^2$$

Option IV (only for smooth loss functions):

Same as Options II but replace $s_{k,i}$ as $s_{k,i} = \dfrac{\lambda\gamma n}{\lambda\gamma n + scl}$

</div>

Figure 1: The Basic Variant of the DisDCA Algorithm (more options)

Taking expectation over $i \in \mathcal{I}_m^k$, we have

$$
\begin{aligned}
\mathrm{E}\left[\frac{A-B}{s}\right] &\geq \sum_{k=1}^{K}\frac{m}{n_k}\sum_{i=1}^{n_k}\left[\phi_{k,i}(x_{k,i}^\top w^{t-1}) + \phi_{k,i}^*(-\alpha_{k,i}^{t-1}) + \alpha_{k,i}^{t-1}x_{k,i}^\top w^{t-1}\right] \\
&\quad - \frac{s}{2\lambda}\left(\frac{mK}{n} - \frac{\gamma(1-s)\lambda}{s}\right)\sum_{k=1}^{K}\frac{m}{n_k}\sum_{i=1}^{n_k}(u_{k,i}^{t-1} - \alpha_{k,i}^{t-1})^2 \\
&= \frac{mK}{n}\sum_{k=1}^{K}\sum_{i=1}^{n_k}\left[\phi_{k,i}(x_{k,i}^\top w^{t-1}) + \phi_{k,i}^*(-\alpha_{k,i}^{t-1}) + \alpha_{k,i}^{t-1}x_{k,i}^\top w^{t-1}\right] \\
&\quad - \frac{s(mK)^2}{2\lambda n}\underbrace{\left(1 - \frac{\gamma(1-s)\lambda n}{smK}\right)\frac{1}{n}\sum_{k=1}^{K}\sum_{i=1}^{n_k}(u_{k,i}^{t-1} - \alpha_{k,i}^{t-1})^2}_{G^t}
\end{aligned}
$$

Note that with $w^t = \nabla g^*(v^t)$, we have $g(w^t) + g^*(v^t) = (w^t)^\top v^t$ and

$$
\begin{aligned}
P(w^{t-1}) - D(\alpha^{t-1}) &= \frac{1}{n}\sum_{k=1}^{K}\sum_{i=1}^{n_k}\phi_{k,i}(x_{k,i}^\top w^{t-1}) + \lambda g(w^{t-1}) - \left[\frac{1}{n}\sum_{k=1}^{K}\sum_{i=1}^{n_k}-\phi_{k,i}^*(-\alpha_{k,i}^{t-1}) - \lambda g^*(v^{t-1})\right] \\
&= \frac{1}{n}\sum_{k=1}^{K}\sum_{i=1}^{n_k}\left(\phi_{k,i}(x_{k,i}^\top w^{t-1}) + \phi_{k,i}^*(-\alpha_{k,i}^{t-1})\right) + \lambda(w^{t-1})^\top v^{t-1} \\
&= \frac{1}{n}\sum_{k=1}^{K}\sum_{i=1}^{n_k}\left(\phi_{k,i}(x_{k,i}^\top w^{t-1}) + \phi_{k,i}^*(-\alpha_{k,i}^{t-1}) + \alpha_{k,i}^{t-1}x_{k,i}^\top w^{t-1}\right)
\end{aligned}
$$

Therefore, we have

$$\mathrm{E}[D(\alpha^t) - D(\alpha^{t-1})] = \mathrm{E}\left[\frac{A-B}{n}\right] \geq \frac{smK}{n}(P(w^{t-1}) - D(\alpha^{t-1})) - \frac{s^2(mK)^2}{2\lambda n^2}G^t$$

$\square$

## 1.1 Proof of Theorem 1

We first prove the case of smooth loss function. We can apply Lemma 1 with $s = \frac{\lambda\gamma n}{\lambda\gamma n + mK}$. In this case, $G^t = 0$. Then we have

$$E[D(\alpha^t) - D(\alpha^{t-1})] \geq \frac{smK}{n} E\left[P(w^{t-1}) - D(\alpha^{t-1})\right]$$

Since $\epsilon_D^{(t-1)} = D(\alpha_*) - D(\alpha^{t-1}) \leq P(w^{t-1}) - D(\alpha^{t-1})$, and $D(\alpha^t) - D(\alpha^{t-1}) = \epsilon_D^{t-1} - \epsilon_D^t$, we obtain

$$E[\epsilon_D^{(t)}] \leq \left(1 - \frac{smK}{n}\right) E[\epsilon_D^{(t-1)}] \leq \left(1 - \frac{smK}{n}\right)^t E[\epsilon_D^{(0)}] \leq \left(1 - \frac{smK}{n}\right)^t \leq \exp\left(-\frac{smKt}{n}\right)$$

$$= \exp\left(-\frac{\lambda\gamma mKt}{\lambda\gamma n + mK}\right)$$

where we use that fact $\epsilon_D^{(0)} = D(\alpha_*) - D(0) \leq 1$, due to that $D(0) \geq 0$, $D(\alpha) \leq P(w^*) \leq P(0) \leq 1$. It is then easy to check that in order to have $E[\epsilon_D^{(T)}] \leq \epsilon_D$, it suffices to have

$$T \geq \left(\frac{n}{mK} + \frac{1}{\lambda\gamma}\right) \log(1/\epsilon_D)$$

Furthermore,

$$E[P(w^t) - D(\alpha^t)] \leq \frac{n}{smK} E(\epsilon_D^{(t)} - \epsilon_D^{(t+1)})] \leq \frac{n}{smK} E[\epsilon_D^{(t)}], \text{ and therefore}$$

$$\sum_{t=T_0}^{T-1} E[P(w^t) - D(\alpha^t)] \leq \frac{n}{smK} E[\left(\epsilon_D^{(T_0)} - \epsilon_D^{(T)}\right)] \leq \frac{n}{smK} E[\epsilon_D^{(T_0)}]$$

We can complete the proof of Theorem 1 for the smooth loss function by plugging the values of $T$ and $T_0$.

## 1.2 Proof of Theorem 2

Next, we prove the case of $L$-Lipschitz continuous loss function. We let $\gamma = 0$ in Lemma 1, and obtain

$$E[D(\alpha^t) - D(\alpha^{t-1})] \geq \frac{smK}{n} E[P(w^{t-1}) - D(\alpha^{t-1})] - \frac{(smK)^2}{2n^2\lambda} G^t$$

where $G^t$ is equal to

$$G^t = \frac{1}{n} \sum_{i=1}^{n} (u_i^{t-1} - \alpha_i^{t-1})^2$$

Following Lemma 4 in , we can bound $G^t$ by $G^t \leq 4L^2$. Let $G = \max_t G^t$. We have

$$E[D(\alpha^t) - D(\alpha^{t-1})] \geq \frac{smK}{n} E[P(w^{t-1}) - D(\alpha^{t-1})] - \left(\frac{smK}{n}\right)^2 \frac{G}{2\lambda}$$

Similarly, the inequality above indicates the following inequality,

$$E[\epsilon_D^{(t)}] \leq \left(1 - \frac{smK}{n}\right) E[\epsilon_D^{(t-1)}] + \left(\frac{smK}{n}\right)^2 \frac{G}{2\lambda} \tag{1}$$

We prove similarly as  the following inequality holds.

$$\epsilon_D^{(t)} \leq \frac{2G}{\lambda(2n/(mK) + t - t_0)} \tag{2}$$

for all $t \geq t_0 = \max\left(0, \left\lceil n/(mK) \log\left(2\lambda n\epsilon_D^{(0)}/(GmK)\right)\right\rceil\right)$. In order to have $\epsilon_D^t \leq \epsilon_D$, it suffice to have

$$T \geq \frac{2G}{\lambda\epsilon_D} + t_0 - \frac{2n}{mK}$$

Furthermore, we have

$$\mathrm{E}[P(\bar{w}^T) - D(\bar{\alpha}^T)] \le \frac{n}{smK(T-T_0)}\mathrm{E}[D(\alpha_*) - D(\alpha^{T_0})] + \frac{G}{2\lambda(T-T_0)} \le \frac{2G}{\lambda(T_0+1)} + \frac{GsmK}{2\lambda n}$$

where $\bar{w}^T = \sum_{t=T_0}^{T_1} w^t/(T-T_0)$ and $\bar{\alpha}^T = \sum_{t=T_0}^{T} \alpha^t/(T-T_0)$. If $T \ge T_0 + \frac{n}{mK}$, we can set $s = n/(mK(T-T_0))$ and obtain

$$\mathrm{E}[P(\bar{w}^T) - D(\bar{\alpha}^T)] \le \mathrm{E}[D(\alpha_*) - D(\alpha^{T_0})] + \frac{G}{2\lambda(T-T_0)} \le \frac{2G}{\lambda(2n/(mK) - t_0 + T_0)} + \frac{G}{2\lambda(T-T_0)}$$

In order to have $P(\bar{w}^T) - D(\bar{b}a^T) \le \epsilon_P$, it suffice to have

$$T_0 \ge \frac{4G}{\lambda\epsilon_P} - \frac{2n}{mK} + t_0, \quad \text{and} \quad T \ge T_0 + \frac{G}{\lambda\epsilon_P}$$

## 2 Derivations of Subproblems in DisDCA and ADMM

We consider the $\ell_2$ regularization where $g(v) = \frac{1}{2}\|v\|_2^2$, $w = v$. Then at $t$-th each iteration, we aim to maximize the dual objective on the sampled data points, i.e.,

$$\max_{\alpha^t} \quad \sum_{k=1}^{K} \sum_{i \in \mathcal{I}_m^k} -\phi_{k,i}^*(-\alpha_{k,i}^t) - \frac{\lambda n}{2}\|w^t\|_2^2$$

$$= \sum_{k=1}^{K} \sum_{i \in \mathcal{I}_m^k} -\phi_{k,i}^*(-\alpha_{k,i}^t) - \frac{\lambda n}{2}\left\| w^{t-1} + \frac{1}{\lambda n}\sum_{k=1}^{K}\sum_{i \in \mathcal{I}_m^k} \Delta\alpha_{k,i}^t x_{k,i} \right\|_2^2$$

$$\ge \sum_{k=1}^{K} \sum_{i \in \mathcal{I}_m^k} -\phi_{k,i}^*(-\alpha_{k,i}^t) - \frac{\lambda n_k}{2}\sum_{k=1}^{K}\left\| w^{t-1} + \frac{1}{\lambda n_k}\sum_{i \in \mathcal{I}_m^k} \Delta\alpha_{k,i}^t x_{k,i} \right\|_2^2$$

$$= \sum_{k=1}^{K} \left( \sum_{i \in \mathcal{I}_m^k} -\phi_{k,i}^*(-\alpha_{k,i}^t) - \frac{\lambda n_k}{2}\left\| w^{t-1} + \frac{1}{\lambda n_k}\sum_{i \in \mathcal{I}_m^k} \Delta\alpha_{k,i}^t x_{k,i} \right\|_2^2 \right)$$

$$= \sum_{k=1}^{K} \left( \sum_{i \in \mathcal{I}_m^k} -\phi_{k,i}^*(-\alpha_{k,i}^t) - \frac{\lambda n_k}{2}\left\| \hat{w}_k^{t-1} + \frac{1}{\lambda n_k}\sum_{i \in \mathcal{I}_m^k} \alpha_{k,i}^t x_{k,i} \right\|_2^2 \right)$$

where $\hat{w}_k^{t-1} = w^{t-1} - \frac{1}{\lambda n_k}\sum_{i \in \mathcal{I}_m^k} \alpha_{k,i}^{t-1} x_{k,i}$. Therefore in each machine the goal of each update is to maximize the following

$$\max_{\alpha} \quad \frac{1}{n_k}\sum_{i=1}^{m} -\phi_i^*(-\alpha_i^t) - \frac{\lambda}{2}\left\| \hat{w}^{t-1} + \frac{1}{\lambda n_k}\sum_{i=1}^{m} \alpha_i^t x_i \right\|_2^2$$

where we suppress the subscript $k$. The above problem has the following primal problem:

$$\min_{w} \frac{1}{n_k}\sum_{i=1}^{m} \phi_i(x_i^\top w) + \frac{\lambda}{2}\left\| w - \left( w^{t-1} - \frac{1}{\lambda n_k}\sum_{i=1}^{m} \alpha_i^{t-1} x_i \right) \right\|_2^2$$

To see this, we have

$$\min_{w} \frac{1}{n_k}\sum_{i=1}^{m} \phi_i(x_i^\top w) + \frac{\lambda}{2}\left\| w - \left( w^{t-1} - \frac{1}{\lambda n_k}\sum_{i=1}^{m} \alpha_i^{t-1} x_i \right) \right\|_2^2$$

$$= \min_{w} \frac{1}{n_k}\sum_{i=1}^{m} \max_{\alpha_i} -x_i^\top w\alpha_i - \phi_i^*(-\alpha_i) + \frac{\lambda}{2}\left\| w - \left( w^{t-1} - \frac{1}{\lambda n_k}\sum_{i=1}^{m} \alpha_i^{t-1} x_i \right) \right\|_2^2$$

$$= \max_{\alpha} \min_{w} \frac{1}{n_k}\sum_{i=1}^{m} -x_i^\top w\alpha_i - \phi_i^*(-\alpha_i) + \frac{\lambda}{2}\left\| w - \left( w^{t-1} - \frac{1}{\lambda n_k}\sum_{i=1}^{m} \alpha_i^{t-1} x_i \right) \right\|_2^2$$

(a) running time (s) vs $m$       (b) DisDCA vs ADMM

We can first compute the minimization to obtain $w = w^{t-1} - \frac{1}{\lambda n_k} \sum_i \alpha_i^{t-1} x_i + \frac{1}{\lambda n_k} \alpha_i x_i$ and then we plug this into above problem, yielding

$$\max_\alpha \quad \frac{1}{n_k} \sum_{i=1}^m -\phi_i^*(-\alpha_i) - \frac{\lambda}{2} \left\| \hat{w}^{t-1} + \frac{1}{\lambda n_k} \sum_{i=1}^m \alpha_i x_i \right\|_2^2$$

In ADMM, the primal objective is decomposed across the examples by imposing equality constraints

$$\max_{w_1,\ldots,w_k,w} \quad \frac{1}{n} \sum_{k=1}^K \sum_{i=1}^{n_k} \phi_{k,i}(w_k^\top x_{k,i}) + \frac{\lambda}{2} \|w\|_2^2$$

$$s.t. \quad w_1 = \ldots = w$$

To solve the problem, a Lagrangian function is constructed

$$L(w_1,\ldots,w_k,w,u_1,\ldots,u_k) = \frac{1}{n} \sum_{k=1}^K \sum_{i=1}^{n_k} \phi_{k,i}(w_k^\top x_{k,i}) + \frac{\lambda}{2} \|w\|_2^2 + \rho \sum_{k=1}^K u_k^\top (w_k - w) + \frac{\rho}{2} \sum_{k=1}^K \|w_k - w\|_2^2$$

Then $\{w_k\}, w, u$ are optimized alternatively by

$$w_k^t = \arg\min_{w_k} \frac{1}{n} \sum_{i=1}^{n_k} \phi_{k,i}(w_k^\top x_{k,i}) + \frac{\rho}{2} \|w_k - (w^{t-1} - u_k^{t-1})\|_2^2$$

$$w^t = \arg\min_w \frac{\lambda}{2} \|w\|_2^2 + \rho \sum_{k=1}^K u_k^\top (w_k - w) + \frac{\rho}{2} \sum_{k=1}^K \|w_k - w\|_2^2$$

$$u_k^t = u_k^{t-1} + w_k^t - w^t$$

## 3   More Experimental Results

We show more experiments in this section. Figure 2(a) presents the running time (for obtaining a duality gap less than $0.01$) versus the values of $m$ for different $\lambda$ and fixed $K = 5$. The results clearly demonstrate the tradeoff incurred by $m$, and also that the effective region of $m$ becomes smaller as $\lambda$ becomes smaller. Figure 2(b) compares the practical variant of DisDCA vs ADMM with different penalty parameters. Figure 2 show more experiments by varying $m$ and $K$ under different settings. Figure 4 compares the different variants of DisDCA.

Figure 2: (a∼d): duality gap with different $m$

Figure 3: duality gap with different $K$

(a) Different Variants

(b) Different Variants

(c) Different Variants

(d) Different Variants

(e) Different Variants

(f) Different Variants

(g) Different Variants

(h) Different Variants

Figure 4: comparison of different variants of DisDCA with different $m$ and $\lambda$ on kdd data.