[Reviews · NeurIPS 2013]

Submitted by Assigned_Reviewer_4

The paper propose a new parallel dual coordinate descent method for solving regularized risk minimization problems. In the proposed algorithm, each machine/core updates a subset of dual variables using the current parameter, and then conduct the "delayed" update to the shared parameter.

The similar idea has been used in yahoo-LDA and matrix factorization (Hogwild), but to my knowledge this is the first applying to dual coordinate descent method. Nice Theoretical guarantee is provided in Theorem 1. Experiments show that the method outperforms other parallel algorithms on linear SVM problems. My only complain is that Figure 3 is too small and hard to read; also, I suggest to use log-scale on primal obj and duality gap.
Summary: This is a good paper and I vote for acceptance.

Submitted by Assigned_Reviewer_5

This paper combines a recent advance in stochastic dual coordinate ascent with the standard approach to parallelize training which is to assign a mini-batch of example to each process and average the resulting gradients. While this combination is novel, its most original contribution is the trade-off bounds between communication and computation. This is likely to be an influential work, introducing a new methodology. The main limitation to its significance are poorly designed experiments. In particular, the authors do not discuss the impact of the input dimension in the bounds, in particular in the case where the data is sparse, which is nearly always the case for big data.

2 major limitations appear in the significance of this trade-off:
- In the discussion on page 5, the authors omit the number of dimensions d. If the data is dense, this could be treated as a constant factor as both the communication and computation cost are linear in (I assume the main loop in computation is W.x which is in O(d)). However the situation becomes very different when the data is sparse: the computation cost for a single dot product scales as the number of non-zero features we denote d’. Thus the total computation cost per iteration is O(m*d’). In the worst case, where no features from the m examples overlap, the communication cost will also be O(m*d’). We then lose this m ratio between them that was critical to the analysis on page 5. Note that in the KDD cup data, one feature out of 1 million is non zero.
- Experiments only count the number of iterations, not the total running time that includes the communication latency. On top of that, the time per iteration depends on the parameters. So experiments reported on figure 3 that show that increasing m always reduce the number of iteration are very misleading, as each single iteration scales as O(m*d) in computation cost, and O(d) in communication cost. A curve showing the total training time, with values for m and K for which this time is minimized would be much more convincing.
In summary, the experiments are so far of little significance, as they only show a reduction of the number of iterations as a function of m and K, which is a trivial observation that does not need theorem 1. Note also that the authors only say they use openMPI, which does say anything about the architecture:
- Cluster: how many nodes, how many core per node?
- Single multicore? Shared memory would mean no communication costs.

The most interesting observation from theorem 1 is the presence of an “effective region” for K and m. But the only thing the experiment show is that decreasing lambda gives more room to choose m and K. A effective upper threshold of the mK product, supported by actual training times, would be a very significant result.

Detailed comments:
Tens of hundreds of CPU cores: do you mean thousands of cores of tens of clusters with hundreds of cores? If communication costs are involved, the target should be clusters, not multicores with shared memory.

Proof of theorem 1: while the proof is correct and trivially obtained from the bound E(epsilon_D^t), the rest of the proof in not needed. In particular, the last sentence is confusing, as there is no T0.
“We can complete the proof of Theorem 1 for the smooth loss function by plugging the values of T and T0.”
There seems to be several lines that have nothing to do with the proof??

Why do the authors repeat twice DisDCA is parameter free? The choice of lambda is critical.

P3, l128: I thought alpha was the associated dual of w, not x, but this may be terminology.
Figure 3: varing->varying.
Plots in Figure 3 are very hard to read.
Summary: An very interesting new algorithm with a theoretical derivation of a communication/computation trade-off: too bad the experiments do not properly demonstrate the trade-off.

Submitted by Assigned_Reviewer_6

This paper investigates a distributed implementation of stochastic dual coordinate ascent (SDCA). The DisDCA algorithm is shown to enjoy the same convergence guarrantees as SDCA, and compared to ADMM it has no parameters that need tuning. Empirically it performs as well as a well-tuned ADMM implementation.

The paper is well-written and technically sound. The algorithm is novel but straightforward and I really enjoyed the section on the tradeoffs between computation and communication. Distributed learning is a very active topic yet not many papers try to analyze the regime at which the algorithms are competitive. In general I don't have any major complaints with the paper except that the figures, both in the main paper, as well as the supplementary material are extremely hard to read. I suggest that Figure 3 contains a sample of the results and the rest are over *multiple pages* in the supplementary material.
Summary: This is a solid paper and relevant to the NIPS community. Distributed DCA is a nice alternative to ADMM.
Author Feedback

Author rebuttal: We are grateful to all reviewers for their useful comments and suggestions. For the common concern, we will make the Figure 3 larger and clearer in the final version.


Review 2 (Assigned_Reviewer_5):

Q: In the discussion on page 5, the authors omit the number of dimensions d. … In the worst case, where no features from the m examples overlap, the communication cost will also be O(m*d’).

A: Thanks for the comments. We would emphasize in the final version that the discussions on Page 5 focus more on dense cases. We would like to mention that the computation and communication tradeoff for the practical variant of DisDCA, which is still an open problem to us, is more involved compared to that of the basic variant. We believe that the convergence rate of the practical variant should have a better dependence on $m$ compared to that of the basic variant as established in Theorem 1, which is indicated by results shown in Figure 3 (c.f. Figure 3c and 3d). As a result, the loss of the $m$ ratio between the computation cost and the communication cost for the basic variant might not carry to the practical variant.

Q: Experiments only count the number of iterations, not the total running time that includes the communication latency

A: We have reported a running time result in the paper (c.f. last several sentences in the paragraph of Tradeoff between Communication and Computation in Section 4). For the deployed setting, the running time of DisDCA-p with m = 1; 10; 100; 1000 by fixing K = 10 are 30; 4; 0; 5 seconds, which implies the trend of running time by increasing m. Similarly, the running time with K = 1; 5; 10; 20 by fixing m = 100 are 3; 0; 0; 1 seconds, respectively.

Q: Note also that the authors only say they use openMPI, which does not say anything about the architecture.

A: We use cluster architecture in our experiments. The largest number of machines used in the experiments is K=20. For each machine, we use one core.

Q: Tens of hundreds of CPU cores: do you mean thousands of cores of tens of clusters with hundreds of cores? If communication costs are involved, the target should be clusters, not multicores with shared memory.

A: the experiments in this paper are performed in a cluster (with a maximum of 20 nodes), although the algorithm can also be employed in a single-node multi-core environment.

Q: Proof of theorem 1: while the proof is correct and trivially obtained from the bound E(epsilon_D^t), the rest of the proof in not needed. In particular, the last sentence is confusing, as there is no T0.

A: Thanks for the proofreading. The last several lines in the proof of Theorem 1 are originally for proving the convergence of the averaged solution. We will remove those.

Q: Why do the authors repeat twice DisDCA is parameter free? The choice of lambda is critical.

A: We will remove one. We would like to mention that the user does not need to specify any parameters (such as e.g., the initial step size in SGD or the penalty parameter in ADMM), except for \lambda, the regularization parameter associated with the problem.